# A Comparative Analysis of a Self-Reported Adverse Events Analysis after Receiving One of the Available SARS-CoV-2 Vaccine Schemes in Ecuador

**DOI:** 10.3390/vaccines10071047

**Published:** 2022-06-30

**Authors:** Esteban Ortiz-Prado, Juan S. Izquierdo-Condoy, Raul Fernandez-Naranjo, Katherine Simbaña-Rivera, Jorge Vásconez-González, Eddy P. Lincango Naranjo, Simone Cordovez, Barbara Coronel, Karen Delgado-Moreira, Ruth Jimbo-Sotomayor

**Affiliations:** 1One Health Research Group, Faculty of Health Science, Universidad de Las Américas, Quito 170507, Ecuador; juan1izquierdo11@gmail.com (J.S.I.-C.); raul.fernandez.n@gmail.com (R.F.-N.); katherine.simbana@msp.gob.ec (K.S.-R.); jorge.vasconez@udla.edu.ec (J.V.-G.); pierina.cordovez@udla.edu.ec (S.C.); barbara.coronel@udla.edu.ec (B.C.); karen.delgado@udla.edu.ec (K.D.-M.); 2Health Management and Research Area, Universidad Internacional Iberoamericana, Campeche 24560, Mexico; 3Facultad de Ciencias Médicas, Universidad Central del Ecuador, Quito 100201, Ecuador; lincane@ccf.org; 4Centro de Investigación para la Salud en América Latina (CISeAL), Pontificia Universidad Católica del Ecuador, Quito 17012184, Ecuador; rejimbo@puce.edu.ec

**Keywords:** COVID-19, vaccines, adverse events, self-reporting, SARS-CoV-2, pandemic

## Abstract

The COVID-19 pandemic has put a lot of pressure on health systems worldwide. Mass vaccination against SARS-CoV-2 has reduced morbidity and mortality worldwide. Despite their safety profiles, vaccines, as with any other medical product, can cause adverse events. Yet, in countries with poor epidemiological surveillance and monitoring systems, reporting vaccine-related adverse events is a challenge. The objective of this study was to describe self-reported vaccine adverse events after receiving one of the available COVID-19 vaccine schemes in Ecuador. A cross-sectional analysis based on an online, self-reported, 32-item questionnaire was conducted in Ecuador from 1 April to 15 July 2021. Participants were invited by social media, radio, and TV to voluntarily participate in our study. A total of 6654 participants were included in this study. Furthermore, 38.2% of the participants reported having at least one comorbidity. Patients received AstraZeneca, Pfizer, and Sinovac vaccines, and these were distributed 38.4%, 31.1%, and 30.5%, respectively. Overall, pain or swelling at the injection site 17.2% (*n* = 4500) and headache 13.3% (*n* = 3502) were the most reported adverse events. Women addressed events supposedly attributable to vaccination or immunization [ESAVIs] (66.7%), more often than men (33.2%). After receiving the first dose of any available COVID-19 vaccine, a total of 19,501 self-reported ESAVIs were informed (87.0% were mild, 11.5% moderate, and 1.5% severe). In terms of the vaccine type and brand, the most reactogenic vaccine was AstraZeneca with 57.8%, followed by Pfizer (24.9%) and Sinovac (17.3%). After the second dose, 6776 self-reported ESAVIs were reported (87.1% mild, 10.9% moderate, and 2.1% severe). AstraZeneca vaccine users reported a higher proportion of ESAVIs (72.2%) in comparison to Pfizer/BioNTech (15.9%) and Sinovac Vaccine (11.9%). Swelling at the injection site, headache, muscle pain, and fatigue were the most common ESAVIs for the first as well as second doses. In conclusion, most ESAVIs were mild. AstraZeneca users were more likely to report adverse events. Participants without a history of COVID-19 infection, as well as those who received the first dose, were more prone to report ESAVIs.

## 1. Introduction

Vaccination is the most cost-effective public health intervention after water sanitation for the control and management of preventable, contagious, and life-threatening diseases [1]. Vaccines induce artificial immunity to several types of microorganisms, avert the likelihood of acquired infections, and reduce morbidity, disability, and mortality [2]. Vaccine development throughout history has been sluggish; however, in February 2020, as the new SARS-CoV-2 virus started to wreak havoc and increase the rate of deaths globally, science was prompt to create a fast, effective, and safe way to counteract the rapid advance of the SARS-CoV-2 virus [3]. The development of vaccinal strategies has never been as rapid and promising as the one experienced, with the development of more than 110 COVID-19 candidates, thanks to the global economic effort due to the planetary priority [3,4].

Several companies and a few countries took the lead, investing millions of dollars and putting together several promising vaccine candidates in record time [4]. In this sense, by 11 December 2020, the Food and Drug Administration [FDA] issued the first emergency-use authorization for the mRNA-based BNT162b2 vaccine produced by Pfizer/BioNTech (New York, NY, USA) [5]. Weeks later, other vaccines came out, and by the first trimester of 2021, several countries began to massively vaccinate their population [6,7] (Table 1).

The Pfizer Phase III study included over 46,000 participants globally, and the results found 95.0% efficacy in preventing infection 7 days after the second dose administration in participants, with these results being pivotal in the FDA granting an EUA for the vaccine in the US.14 On 31 March 2021, Pfizer and BioNTech announced remarkable 100% vaccine efficacy in young adolescents aged 12–15 years, after phase 1/2/3 trials [6,7].

Followed by other companies, vaccine trials were starting to show communitarian benefits, and several countries started to vaccinate their populations [8,9,10] (Table 1).

In general, regardless of the immunologic mechanism used by any of the COVID-19 vaccines, serious adverse events are very rare, and most of the symptoms associated with the vaccines are mild [11]. For instance, in Poland, one ESAVI was reported for every 1100 vaccines applied [12]. Similarly, Beaty et al. reported that serious ESAVIs were very rare in a cohort of more than 19,000 patients in the U.S. [13].

Most ESAVIs are mild and self-limited and can present as systemic or local reactions. A wide spectrum of ESAVIs has been reported, with pain, redness, or swelling at the vaccine injection site, fever, fatigue, headache, muscle pain, nausea, vomiting, itching, chills, and joint pain being the most common [14,15]. In very rare cases, vaccines can also cause serious adverse reactions such as anaphylactic shock, vaccine-induced immune thrombotic thrombocytopenia [VITT], or even death, with a disputable rate of less than 0.0025% of deaths according to the Centers for Disease Control and Prevention [CDC] [16].

The most serious documented adverse evets are myocarditis (6/millions), transverse myelitis (0.01/100,000), appendicitis (5/100,000), Bell’s palsy (2.6/100,000), and pericarditis (4.9/million); in the case of the CoronaVac vaccine, the ESAVIs were Bell’s palsy (3.8/100,000), Encephalopathy (0.01/100,000), thromboembolic events (0.61–1.15/million), Guillain–Barré syndrome (0.29/million), or thrombosis with thrombocytopenia syndrome (1503 cases among 592 million doses) [17].

The identification and reporting of events supposedly attributable to vaccination or immunization (ESAVI) is a priority [18]. Although COVID-19 vaccines have shown numerous benefits, some people are still wary of COVID-19 vaccines [19]. This study aimed to describe the self-reported vaccine adverse events after the application of the available COVID-19 vaccines in Ecuador.

## 2. Materials and Methods

### 2.1. Study Design and Sample Selection

We conducted a cross-sectional study by circulating a 32-item, self-reported, online questionnaire through the internet-based survey platform Survey Monkey. We gathered anonymous responses from all over the country using a non-probability sampling method from 1 April to 15 July 2021.

### 2.2. Settings and Population

Participants were all Ecuadorian residents who received one of the available COVID-19 vaccines. According to the current government plan, 9 million people were vaccinated by July 2021. In that sense, with a confidence level of 95% and a margin of error of 1.5%, our minimum estimated sample was 4267 responses.

Using the expected population to be vaccinated, the sample size “n” and margin of error “E” are given by the following formula [20,21]:x = Z(c/100)2r(100 − r)
n = N x/((N − 1)E2 + x)
E = Sqrt[(N − n)x/n(N− 1)]
where “N” is the population size, “r” is the fraction of responses (50%), “Z(c/100)” is the critical value for the confidence level “c”, “x” is the expected population, and Sqrt is the square root.

All responses that came from respondents who voluntarily agreed to participate in the study and who completed all 32 questions were included. Moreover, only participants that reported they received the mRNA vaccine (Pfizer/BioNTech), viral vector (AstraZeneca), or inactivated vaccine (Sinovac) were included.

### 2.3. Survey Development and Measures

The data were collected using a 32-item online anonymous questionnaire to evaluate self-reported symptoms and adverse events after receiving one or two doses of the available COVID-19 vaccines. Participants’ consent was obtained at the beginning of the questionnaire with an explanation of the objective of the study. Participants could proceed with the full questionnaire only after obtaining consent by accepting (electronically marking) a ‘Terms and Conditions’ and ‘Participation Agreement’ consent form. A novel questionnaire instrument was developed for this study using items adapted from information about COVID-19 published by the CDC and the World Health Organization [WHO] alongside items used in previous COVID-19 surveys [22,23]. The questionnaire was developed and fielded in Spanish and later translated into English for reporting purposes. The full survey instrument is available in Appendix A.

The questionnaire consisted of four sections: Demographic characteristics, comorbidities, past medical history of COVID-19, and vaccination information. The demographic section collected information on age, sex, occupation, and place of residency. The comorbidities section includes questions about respondents’ medical history not related to COVID-19 (smoking status, medications, and other diseases). The COVID-19-related medical history was composed of 6 questions that include information related to previous infections. Finally, the vaccination information section was composed of 20 questions that evaluated the type, date, place, and number of doses of the administered vaccine. Moreover, it evaluated the timing and type of vaccine-related symptoms and adverse events.

The questionnaire was sent through national broadcasting news and social media, using snowballing as a sample method. Weekly reminders were sent to participants during the period of the questionnaire.

### 2.4. Data Management

We reviewed data case by case to ensure the highest accuracy possible on our results. As such, we identified cases where responses did not match the questions posed. For example, when they answered that they had not received the second dose of the vaccine, but in the section of adverse events attributed to the second dose they chose a response, this type of case was automatically eliminated. Moreover, we excluded responses coming from the same IP address to avoid malicious and duplicate answers.

### 2.5. Statistical Analysis

Descriptive and inferential analysis was conducted using the software IBM SPSS Statistics for Windows Version 24.0. The results of each item in the questionnaire were reported as men and women in percentages and absolute frequencies with no further intersex variability analysis.

The Chi-Square test was used to test the association between nominal qualitative variables. A Chi-Square test for the independence of variables was chosen to search for a relationship between variables such as gender, age, comorbidities, history of COVID-19, type of vaccine, and ESAVIs. The method for calculating the p-value was based on the frequentist approach to the test. For Tables 3, 5 and 6, the data were stratified for each adverse effect reported, and contingency tables were constructed for each group based on the variables mentioned. Finally, *p*-values were obtained according to the definition of the chi-square distribution. In the case of Table 4, the same approach was used; however, the variable “dose number” (first and second doses) was used to verify independence with the following variables: Time of ESAVIs onset, ESAVIs severity (mild, moderate, and severe), and measures to mitigate self-reported ESAVIs.

An independent *t*-test was employed to evaluate the existence of an association between quantitative variables (age and number of ESAVIs). Figure 4 presents a contrast between age groups and the number of ESAVIs. The *t*-test was calculated using the null hypothesis that age level “i” has the same number of adverse effects as age level “i-1”. For age ranges with *p*-values  < 0.05, the null hypothesis was rejected (different number of adverse effects between age groups).

Statistical significance was accepted at *p* < 0.05. Confidence intervals at 95% from means and proportions were also computed.

### 2.6. Reliability and Validation

Reliability was examined using a test–retest questionnaire using the final version of the survey. Since this questionnaire was created only for this project, we tested within the cohort of experts previously selected for informal interviews.

### 2.7. Ethical Approval

This project is part of our COVID-19 analysis program, a nationwide study of the epidemiology of COVID-19 in Ecuador, which received an exemption letter from the UDLA’s IRB named CEISH on 10 March 2020. The current analysis included anonymized, un-identifiable information and it complies with all local and international guidelines regarding the ethical use of anonymous, non-identifiable information stipulated in such documents as the Declaration of Helsinki and the Good Clinical Practice Guidelines (GCP).

## 3. Results

### 3.1. Demographic Information and Past Medical History

A total of 11,927 responses were collected at the national level; however, after revalidation, 6654 responses were considered valid for analysis. From the total responses, 4018 (60.4%) were women, 2631 (39.5%) were men, and 5 (0.1%) defined themselves as others. Most respondents were aged between 21 and 50 years (73.9%) and most of the participants were employed (79.60%); of them, 21.2% were healthcare workers. A total of 4104 participants reported having no comorbidities, and out of the remaining 2547 participants, the most frequent comorbidity was hypothyroidism (24.5%) and chronic diseases such as obesity (19.9%) and arterial hypertension (17.0%). Most of the participants did not have COVID-19 at the time of the questionnaire. The three vaccine types available in Ecuador were similarly distributed (Table 2).

At the time of the survey, for 6654 participants, a higher vaccination rate per 100,000 Ecuadorian inhabits was found in AstraZeneca vaccine users for “younger” participants between 20 and 60 years of age, while the highest vaccination rate in participants older than 60 years was found in the Pfizer vaccine. (Figure 1).

### 3.2. Self-Reported ESAVIs

#### 3.2.1. General Characteristics

Overall, the number of adverse effects reported by participants between the first and second doses was *n* = 26,325, and women had a higher frequency of self-reported ESAVIs 66.7% (*n* = 17,507). Most symptoms were reported by participants who claimed not to have comorbidities 58.0% (*n* = 15,206) and in those lacking a personal history of COVID-19 infection 72.7% (*n* = 19,081) (Table 3).

After the first vaccine dose, a total of 19,501 self-reported ESAVIs were reported; 87.0% were mild, 11.5% were moderate, and 1.5% were severe. For the second dose, there were 6776 self-reported ESAVIs (87.1% mild, 10.9% moderate, and 2.1% severe) (Table 4).

The behavior of the reported effects shows a common trend when analyzed among all different age groups (nine groups) in favor of pain or swelling at the injection site in all age groups as the most reported 17.2% (*n* = 4500); moreover, headache is presented as the second most frequent with 13.3% (*n* = 3502). (Table 5). According to the ESAVIs, for participants between 10 and 40 years and over 70 years of age, the most frequent symptom was fatigue, while for the group between 40 and 70 years old, the most frequent symptom was pain or swelling at the injection site and headache (Table 5).

Self-reported ESAVIs were higher among young adults (21 to 39 years), with 55.8% and adults (40 to 64 years), with 33.2%, compared to older adults (>65 years), with 7.6%; however, there are no significant differences in terms of the occurrence of adverse events after performing the corresponding statistical tests (Appendix A).

Figure 2 shows the effect of the comparison of the number of adverse events between the different age ranges. It was found that the effects due to vaccination differ only between the age ranges up to 70 years (*p* < 0.05). In participants older than 70 years, the number of adverse effects is very similar (*p* > 0.05) (Figure 2). Along the same lines, when addressing the influence of age between only two age groups, taking 50 years as the cut-off point, it was shown that the group under 50 years of age presents various symptoms more frequently (*p* < 0.05) (Table 3).

The analysis of the development of ESAVIs for each age group shows that for participants in the 30 to 40 years age group, a higher number of ESAVIs were reported for AstraZeneca vaccine users compared to Sinovac vaccine (*p* = 0.016) as well as for participants in the 40 to 50 years age group (*p* = 0.028) (Figure 3).

#### 3.2.2. Self-Reported ESAVIs after the First Vaccine Dose

The proportion of ESAVIs after the first vaccine dose was higher in users of AstraZeneca (57.8%), followed by Pfizer (24.9%) and Sinovac (17.3%). In the same way, the most common self-reported ESAVIs with the AstraZeneca vaccine were headache (9.1%), muscle pain (8.9%), and discomfort (8.6%). For the Pfizer/BioNTech vaccine, the most common self-reported ESAVIs were pain or swelling at the injection site (16.5%), headache (7.7%), and fatigue or tiredness (7.6%). Finally, for the Sinovac inactivated virus vaccine, pain or swelling at the injection site (11.3%), fatigue or tiredness (9.8%), and headache (9.7%) were the most common self-reported ESAVIs (Table 6).

#### 3.2.3. Self-Reported ESAVIs after the Second Vaccine Dose

Second-dose AstraZeneca vaccine users reported a higher proportion of ESAVIs (72.21%) compared to Pfizer/BioNTech (15.85%) and Sinovac (11.93%) vaccines. For the AstraZeneca vaccine, headache (8.7%), pain or swelling at the injection site (7.7%), and fatigue or tiredness (7.2%) were the more common self-reported ESAVIs. After the Pfizer/BioNTech vaccine, the most common self-reported ESAVIs were pain or swelling at the injection site (10.3%), headache (7.2%), and muscle pain (6.7%). Finally, with the Sinovac vaccine, participants self-reported pain or swelling at the injection site (9.5%), headache (7.6%), and fatigue or tiredness (6.8%) more frequently (Table 7).

In general, the effect of the available vaccines in Ecuador showed that the most common adverse event was pain at the injection site, followed by headache (Figure 4).

## 4. Discussion

To the best of our knowledge, this is the first study in Latin America examining self-reported ESAVIs after one or two COVID-19 vaccine doses. This analysis revealed that women, young adults, participants without a COVID-19 infection history, participants after the first vaccine dose, and AstraZeneca users developed ESAVIs more frequently. Most self-reported ESAVIs were mild and moderate.

Although this sample only represents a small subset of all vaccinated people in Ecuador, our results are similar to those described in other reports. Regarding sex, we found a higher proportion of female participants, who showed more pain or swelling at the injection site, muscle pain, and swollen glands (*p* < 0.05) compared to the male group and other groups. These results are similar to those seen in several studies [24,25,26]. Despite the differences in the distribution of the participants according to age, in the group younger than 50 years (81.2%) and the group older than 50 years (18.8%), the presentation of self-reported ESAVIs showed significant results (*p* < 0.05) that participants younger than 50 years had more ESAVIs, especially mild and moderate symptoms, in comparison with those older than 50 years. This is in line with data reported by Zare et al. who observed a greater presence of adverse events in those under 40 years of age. Moreover, Cuschieri et al. and Menni et al. found more reports of ESAVIs in people under 45 years of age and 55 years of age, respectively [22,27,28].

One interesting and controversial point is the relationship between COVID-19 history and ESAVIs. In our study, participants without a history of infection at the time of the questionnaire reported more adverse events. Our findings contrast with several studies [22,27]. Regarding these differences, an increase in reactogenicity related to an increase in antibody titers conditioned by a previous infection in vaccinated individuals has been proposed. However, to date, this causal relationship has not been fully demonstrated, assuring us that the role of the history of the previous infection does not cause an increased number of ESAVIs in patients receiving vaccines against the SARS-CoV-2 virus.

The most common reported ESAVIs were pain or swelling at the injection site, headache, and fatigue, as observed by several authors in other investigations based on their own reports, such as Menni et al. in the UK, Elgendy et al. in the Egyptian population, as well as in the clinical trial by Polack et al. [22,24,29]. When comparing the frequency of ESAVIs according to the number of doses, no important differences were found between the ESAVIs reported after the first and second doses, although the absolute number of ESAVIs reported was higher after the first dose (n = 19,501 vs. n = 6776), a finding that is similar to recent reports [29]. In addition, among the vaccines types available in Ecuador, for AstraZeneca and Sinovac, a similar trend was seen with a higher number of ESAVIs reported in the administration of the first dose, while with the Pfizer vaccine, a slight increase in ESAVIs was found in the administration of the second dose. Regarding these discrepancies, previous studies with larger populations showed that the participants studied had a greater number of adverse effects caused by the second dose of the AstraZeneca and Pfizer vaccines [23,24].

Regarding differences in the severity of symptoms, it is interesting that the most frequent moderate ESAVIs reported, in general, were fever and nausea, while within the severe ESAVIs, changes in heart rhythm (tachycardia) occurred in more than half of the reports.

The severe ESAVIs found in our study were very infrequent (overall n < 30). In this group, self-reports of Guillain–Barre Syndrome, transverse myelitis, anaphylaxis, and coagulation disorders were obtained for the three vaccines available in Ecuador (AstraZeneca, Pfizer, Sinovac), in addition, the frequency of presentation of severe ESAVIs was higher after receiving the first dose of vaccines. In the available literature of self-reporting studies, no data have been presented regarding the severe adverse effects reported in our research. Only one self-reporting study developed by Mathioudakis et al. found reports of anaphylaxis [30], the difficulty in accepting the reports of severe adverse effects as valid is likely due to the self-reported nature of the investigations, and, in the case of our study, we had no way of assuring that the diagnoses were made by physicians.

The origin and cause of these findings are difficult to explain, and in the case of the relationship between vaccines against COVID-19 and the development of Guillain–Barre syndrome, an analysis performed by Keh et al. in databases from the United Kingdom stated that there are no demographic or phenotypic differences that can ensure the development of Guillain–Barre syndrome due to COVID-19 vaccines; however, most of the population studied was of the UK white ethnicity (90.3%) with a minimal presence of Latin Americans [26]. Otherwise, in the case of anaphylaxis, an extensive meta-analysis managed to demonstrate that, although reports of anaphylaxis due to the possible cause of SARS-CoV-2 virus vaccines are higher than with other rare adverse effects, these cannot be attributed to the receipt of these vaccines, as they occur with a wide range of vaccines and are mainly related to vaccine components [25].

Regarding the ESAVIs reported by users who received the CoronaVac vaccine (Sinovac) in our study, the most frequently recorded were pain or swelling at the injection site, headache, and fatigue for the first and second doses. In agreement with these data, Zhang et al. found that the most frequent ESAVIs recorded were fatigue, muscle pain, and headache for both doses of Sinovac in a Chinese population, coinciding only with headache in our findings [31]; while in a Turkish population, the findings by Riad et al. were shown to be more similar to what was found in our study, since the most reported ESAVIs were pain at the injection site, fatigue, headache, and muscle pain [32].

Our findings show differences in the frequency of reporting of ESAVIs when comparing the type of vaccine, and the data showed that the greatest number of ESAVIs was attributed to the ChAdOx1 viral vector vaccine, the same as that presented by Menni et al., Klugar et al., Omeish et al., and Alhazmi et al. [22,33,34]. Furthermore, among the 30 to 40 years and 40 to 50 years age groups, participants who received the AstraZeneca vaccine had a higher number of reported ESAVIs compared to users of other vaccines; these differences could be due to the fact that, in participants aged 20 to 60 years, the most used vaccine was ChAdOx1.

Addressing these differences in the number of doses, the reported distribution shows different patterns. In the case of the first dose, the ChAdOx1 vaccine was responsible for the majority of adverse events, while among participants that received the second dose of vaccination, the BioNTech vaccine was the one that was attributed to more than half of ESAVIs, and this trend is similar to that found in Egypt [29].

In the analysis of the time of ESAVIs onset, we found that the majority of ESAVIs significantly occurred within the first day after receiving the vaccine. Other authors observed the same behavior in their participants, although they did not find statistically significant differences [29,34,35]. On the other hand, the measures that the adverse events caused our participants to take were characterized by taking medication to correct the symptoms in most cases, and in a very small proportion, to visit a doctor. These behaviors have been previously observed in similar proportions in the population of Saudi Arabia [34], and almost zero participants required hospitalization due to adverse events. Likewise, this trend was similarly evidenced in studies that evaluated adverse effects due to AstraZeneca, Pfizer vaccines, and Sinopharm [34,35].

## 5. Limitations

Our study has several limitations inherent to the cross-sectional self-reported design. Since the questionnaire was distributed through social networks, the information belonging to the population that did not have the resources to access the questionnaire was relegated from the study. Similarly, most older adults handle electronic devices and the Internet with difficulty. This may cause selection bias; however, we consider that our sample calculation provided a considerable sample size to reduce this bias. Social desirability bias is another potential limitation that could have affected responses due to the self-report nature of the questionnaire. In other words, negative or positive self-reported symptoms may be underreported or overreported because respondents want to mark what they consider “socially acceptable” responses. However, the use of an anonymous online questionnaire should have somewhat mitigated the risk of this bias. Despite these limitations, we managed to obtain a large amount of information provided by many participants based on official data on the number of vaccinated in Ecuador. Likewise, our data were subjected to extensive filtering to provide valid results. For these reasons, we consider this study to be a reliable approximation of the self-reported ESAVIs of Ecuadorians that can help to inform their symptoms and somehow decrease COVID-19 hesitancy.

## 6. Conclusions

This is the first report in Ecuador and the Latin-American region on the adverse events attributed to COVID-19 vaccines. The results of our study allow us to conclude that, in Ecuador, the majority of ESAVIs are mild to moderate and occur more frequently after receiving the AstraZeneca vaccine, followed by Pfizer and finally Sinovac. We also noted that women are more likely to report ESAVIs compared to men and that young people report them more frequently than women. In general, it can be observed that the adverse events reported are mostly mild and transitory, which further supports the fact that COVID-19 vaccines are safe. Mild ESAVIs were the most self-reported events described in Ecuador. We found that first doses were associated with a higher proportion of ESAVIs compared to subjects who received a subsequent dose. We also found that participants without a past medical history of COVID-19, AstraZeneca users, and younger populations are more likely to develop ESAVIs.

## Figures and Tables

**Figure 1 vaccines-10-01047-f001:**
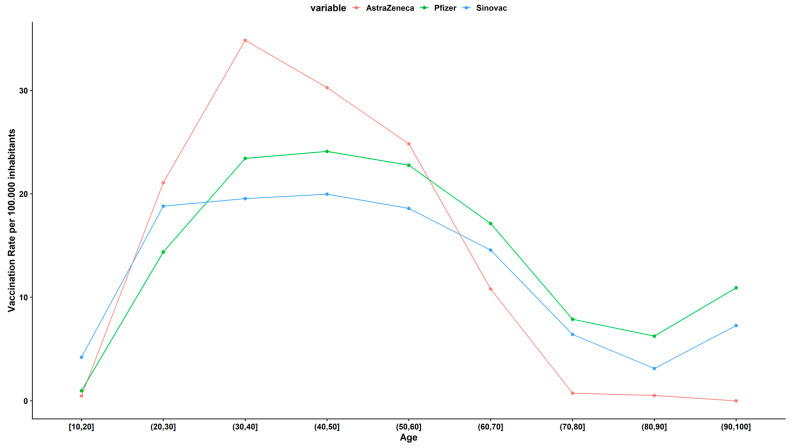
Distribution of COVID-19 vaccines among 6654 participants relative to the Ecuadorian population.

**Figure 2 vaccines-10-01047-f002:**
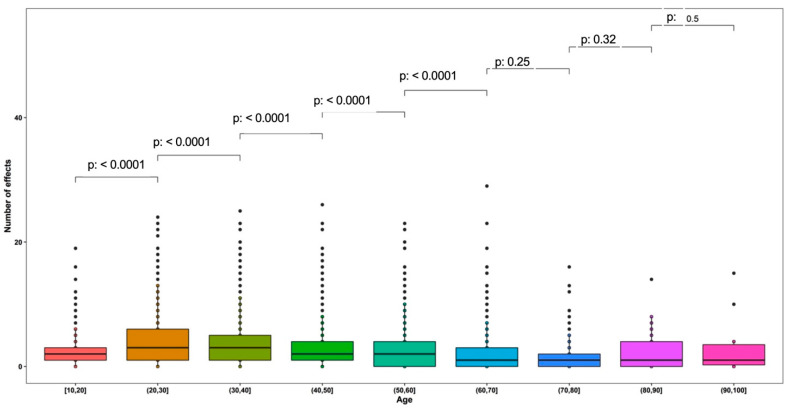
*t*-test analysis among number of ESAVIs reported and age groups.

**Figure 3 vaccines-10-01047-f003:**
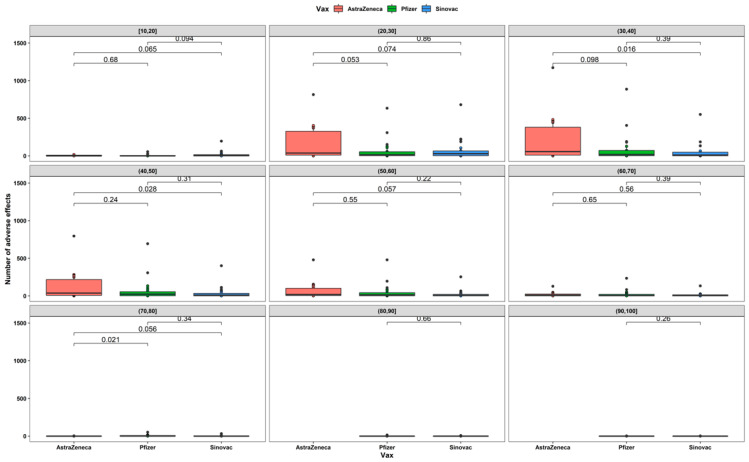
*t*-test between number of ESAVIs and vaccine type by age range.

**Figure 4 vaccines-10-01047-f004:**
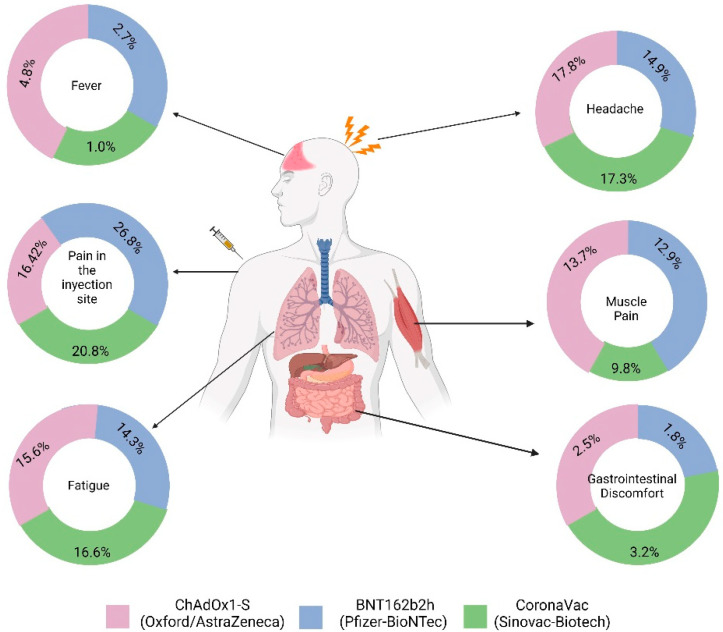
Symptoms depending on the vaccine administered.

**Table 1 vaccines-10-01047-t001:** The main characteristics of the different mechanisms of action used by different COVID-19 vaccines.

Type of Vaccine	Mechanism of Action	Vaccine Name	Producer	Dose	Storage
mRNA	Protein S extension with proline substitutions	BNT162b2	Pfizer/BioNtech	30 µg 2 doses, interval 21 days	−25 °C to −15 ° C; 2–8 °C for 30 days. Ambient temperature ≤ 12 h
Viral vector	Replication-deficient Chimp adenovirus viral vector and SARS-CoV-2 protein S	ChAdOx1 (AZS1222)	AstraZeneca/Oxford	5 × 10^10^ viral particles 2 doses 28-day interval	2–8 °C for 6 months
Viral vector	Human adenovirus serotype 26 viral vector with incompetent recombinant replication and stabilized SARS-CoV-2 protein S	Ad26.COV 2.S	Janssen/Johnson & Johnson	5 × 10^10^ viral particles 1 dose	−20 °C; 2–8 °C for 3 months
mRNA	Spike S protein extension with proline substitutions	mRNA-1273	Modern	100 µg 2 doses 28-day intervals	−25 °C to −15 °C; 2–8 °C for 30 days; ambient temperature ≤ 12 h
Inactivated virus	SARS-CoV-2 Inactivated HB02 Chain Created from Vero Cells	BBIBP-CorV	Sinopharm ½	4 µg with aluminum hydroxide adjuvant 2 doses 21-day interval	2–8 °C duration unknown
Inactivated virus	SARS-CoV-2 Inactivated CN02 Chain Created from Vero Cells	CoronaVac	Sinovac Biotech	3 µg with aluminum hydroxide adjuvant 2 doses 14-day interval	2–8 °C duration unknown
Viral vector	SARS-Cov-2 glycoprotein S full-length loaded adenovirus vector	Gam-COVID-Vac (Sputnik V)	Gamaleya National Research Center for Epidemiology and Microbiology	10^11^ viral particles per dose for each recombinant adenovirus 2 doses 1st rAd26 2nd rAd5 21-day interval	−18 ° C liquid; 2–8 °C frozen dry 6 months
Inactivated virus	SARS-CoV-2 Inactivated NIV-2020-770 Chain Created from Vero Cells	BBV152 (Covaxin)	Bharat Biotech/Indian Council of Medical Research	6 µg of whole-virion inactivated SARS-CoV-2 antigen (Strain: NIV-2020-770) 0.5 mL 2 doses 28-day interval	2–8 °C for 6 months
Protein subunit	Encodes the SARS-CoV-2 RBD antigen (residues 319–537, accession number YP_009724390), with two copies in tandem repeat dimeric form	ZF2001 (Zifivax)	Anhui Zhifei Longcom/Institute of Microbiology at the Chinese Academy of Sciences	25 µg 3 doses (0.1 and 4–6 months)	2–8 °C duration unknown
Viral vector	Recombinant human Adenovirus type 5 viral particlesthat is incapable of replication and which expresses the protein S (spike) of SARS-CoV-2	AD5-nCOV (Convidecia)	CanSino Biologics	4 × 10^10^ recombinant human Adenovirus type 5 viral particles 0.5 mL 1 dose	2–8 °C
Protein subunit	Monomeric receptor binding domain subunit, residues 331–530 of the Spike protein of SARS-CoV-2 strain 156 Wuhan-Hu-1	CIGB-66 (ABDALA)	Center for Genetic Engineering and Biotechnology	0.5 mL 3 doses 14-day interval	2–8 °C for 6 months
Conjugate	SARS-CoV-2 virus receptor binding domain (RBD) recombinant protein monomeric) conjugated to tetanus toxoid	FINLAY-FR-2 (Soberana 02)	Finlay Institute	0.5 mL Two doses 28-day interval. A third (booster) dose of Soberana Plus may also be given on day 56	2–8 °C for 6 months

**Table 2 vaccines-10-01047-t002:** Demographic characteristics of the study population.

Demographic Characteristics			
		n	(%)
**Gender**	Female	4018	60.4%
	Male	2631	39.5%
	Other	5	0.1%
	**Total**	**6654**	**100%**
**Age (years)**	10 to 20	204	3.1%
	21 to 30	1553	23.3%
	31 to 40	1895	28.5%
	41 to 50	1471	22.1%
	51 to 60	997	15.0%
	61 to 70	428	6.4%
	71 to 80	82	1.2%
	81 to 90	19	0.3%
	91 to 100	5	0.1%
	**Total**	**6654**	**100%**
**Occupation**	Health care workers	1405	21.1%
	Outside health care area workers	3892	58.5%
	Unemployed	1357	20.4%
	**Total**	**6654**	**100%**
**Live in Ecuador**	Yes	6634	99.7%
	No	20	0.3%
	**Total**	**6654**	**100%**
**Comorbidities**	Asthma	188	7.4%
	Cardiovascular disease	164	6.4%
	Arterial hypertension	432	17.0%
	Obesity	506	19.9%
	Hypothyroidism	624	24.5%
	Cancer	25	1.0%
	Chronic liver disease	25	1.0%
	Type 1 Diabetes	49	1.9%
	Type 2 Diabetes Mellitus	135	5.3%
	Hyperthyroidism	49	1.5%
	Psoriasis	95	3.7%
	Chronic kidney disease	17	0.7%
	Respiratory disease	61	2.4%
	Stroke (ischemic or hemorrhagic)	16	0.6%
	Chronic neurological disorders	30	1.2%
	Coagulation disorders	59	2.3%
	Tuberculosis	2	0.1%
	Peptic ulcers	57	2.2%
	HIV/AIDS	13	0.5%
	**Total**	**2547**	**100%**
**History of COVID-19 infection**	I think yes, but did not take the test	264	4.0%
	Yes/confirmed	1450	21.8%
	No	4940	74.2%
	**Total**	**6654**	**100%**
**Vaccine type**	Pfizer	2069	31.1%
	AstraZeneca	2553	38.4%
	Sinovac	2032	30.5%
	**Total**	**6654**	**100%**

**Table 3 vaccines-10-01047-t003:** Self-reported adverse events after receiving the first and second dose of any of the available COVID-19 vaccines distributed by Ecuadorian participants characteristics.

		Symptoms	Gender n (%)				Comorbidities n (%)			History of COVID-19 Infection n (%)			Age n (%)		
	Total n		Female	Male	Other	*p* Value *	No	Yes	*p* Value *	No	Yes	*p* Value *	<50 Years	>50 Years	*p* Value *
**Mild self-reported ESAVIs**	22,814	Joint pain	1083 (66.9)	535 (33.0)	1 (0.1)	0.200	890 (55.0)	729 (45.0)	**<0.001**	1151 (71.1)	468 (28.9)	0.799	1292 (79.2)	327 (20.2)	**<0.001**
		Headache	2355 (67.2)	1143 (32.6)	4 (0.1)	0.593	2021 (57.7)	1481 (42.3)	**<0.001**	2565 (73.2)	937 (26.8)	**<0.001**	2785 (79.5)	717 (20.5)	**<0.001**
		Muscle pain	1865 (64.4)	1028 (35.5)	4 (0.1)	**<0.001**	1683 (58.1)	1214 (41.9)	**<0.001**	2086 (72.0)	811 (28.0)	0.708	2308 (79.7)	589 (20.3)	**<0.001**
		Pain or swelling at the injection site	2969 (66.0)	1528 (34.0)	3 (0.1)	**0.045**	2570 (57.1)	1930 (42.9)	**<0.001**	3337 (74.2)	1,163 (25.8)	**<0.001**	3609 (80.2)	891 (19.8)	**<0.001**
		Fatigue or tiredness	2209 (66.9)	1088 (32.9)	5 (0.2)	0.192	1931 (58.5)	1371 (41.5)	**<0.001**	2480 (75.1)	822 (24.9)	0.056	2739 (82.9)	563 (17.1)	**<0.001**
		Shaking chills	1375 (65.5)	723 (34.4)	2 (0.1)	0.071	1302 (62.0)	798 (38.0)	**<0.001**	1506 (71.7)	594 (28.3)	0.582	1760 (83.8)	340 (16.2)	**<0.001**
		Mild/low grade fever (37.1–38 °C)	941 (64.1)	527 (35.9)	1 (0.1)	0.385	889 (60.5)	580 (39.5)	**<0.001**	1005 (68.4)	464 (31.6)	0.374	1245 (84.8)	224 (15.2)	**<0.001**
		General Discomfort	1801 (64.5)	988 (35.4)	5 (0.2)	0.691	1692 (60.6)	1,102 (39.4)	**<0.001**	2005 (71.8)	789 (28.2)	0.283	2300 (82.3)	494 (17.7)	**<0.001**
		Gastrointestinal Discomfort	367 (69.4)	162 (30.6)	0 (0.0)	0.328	286 (54.1)	243 (45.9)	**<0.001**	405 (76.6)	124 (23.4)	**0.023**	419 (79.2)	110 (20.8)	0.075
		Urinary Discomfort	71 (69.6)	31 (30.4)	0 (0.0)	0.288	67 (65.7)	35 (34.3)	0.086	68 (66.7)	34 (33.3)	0.225	85 (83.3)	17 (16.7)	0.297
**Moderate self-reported ESAVIs**	2987	Diarrhea	287 (71.0)	117 (29.0)	0 (0.0)	1	213 (52.7)	191 (47.3)	**<** **0.001**	305 (75.5)	99 (24.5)	**0.004**	308 (76.2)	96 (23.8)	**0.004**
		Skin rash	116 (73.0)	43 (27.0)	0 (0.0)	0.776	82 (51.6)	77 (48.4)	**0.0166**	119 (74.8)	40 (25.2)	**0.019**	124 (78.0)	35 (22.0)	**<0.001**
		Fever (>38 °C)	536 (64.7)	291 (35.1)	2 (0.2)	0.857	523 (63.1)	306 (36.9)	**<0.001**	578 (69.7)	251 (30.3)	**0.006**	686 (82.8)	143 (17.2)	**<0.001**
		Swollen Glands	315 (74.3)	108 (25.5)	1 (0.2)	**0.048**	235 (55.4)	189 (44.6)	**<0.001**	308 (72.6)	116 (27.4)	0.844	351 (82.8)	73 (17.2)	**0.045**
		Nausea	564 (77.2)	166 (22.7)	1 (0.1)	0.389	406 (55.5)	325 (44.5)	**<0.001**	545 (74.6)	186 (25.4)	0.843	606 (82.9)	125 (17.1)	0.544
		Pruritus (Itching)	239 (79.4)	62 (20.6)	0 (0.0)	0.833	158 (52.5)	143 (47.5)	**0.019**	206 (68.4)	95 (31.6)	1	249 (82.7)	52 (17.3)	**<0.001**
		Vomit	110 (79.1)	29 (20.9)	0 (0.0)	0.808	75 (54.0)	64 (46.0)	**<0.001**	99 (71.2)	40 (28.8)	**0.022**	115 (82.7)	24 (17.3)	0.056
**Severe self-reported ESAVIs**	434	Anaphylaxis	4 (44.4)	5 (55.6)	0 (0.0)	1	5 (55.6)	4 (44.4)	0.906	5 (55.6)	4 (44.4)	0.906	8 (88.9)	1 (11.1)	0.495
		Tachycardia	191 (74.3)	66 (25.7)	0 (0.0)	0.830	112 (43.6)	145 (56.4)	**0.003**	189 (73.5)	68 (26.5)	0.702	193 (75.1)	64 (24.9)	0.398
		Guillain Barre sd.	4 (40.0)	6 (60.0)	0 (0.0)	1	9 (90.0)	1 (10.0)	1	9 (90.0)	1 (10.0)	1	8 (80.0)	2 (20.0)	1
		Facial swelling	17 (65.4)	9 (34.6)	0 (0.0)	1	12 (46.2)	14 (53.8)	0.254	18 (69.2)	8 (30.8)	1	18 (69.2)	8 (30.8)	0.279
		High Blood Pressure	62 (71.3)	25 (28.7)	0 (0.0)	0.227	21 (24.1)	66 (75.9)	**0.002**	66 (75.9)	21 (24.1)	0.529	50 (57.5)	37 (42.5)	0.922
		Transverse Myelitis	1 (50.0)	1 (50.0)	0 (0.0)	N/D	1 (50.0)	1 (50.0)	N/D	1 (50.0)	1 (50.0)	#N/D	2 (100.0)	0 (0.0)	N/D
		Facial Paralysis	7 (63.6)	4 (36.4)	0 (0.0)	0.206	6 (54.5)	5 (45.5)	0.206	8 (72.7)	3 (27.3)	1	8 (72.7)	3 (27.3)	0.513
		Coagulation Disorders	18 (56.3)	14 (43.8)	0 (0.0)	0.355	17 (53.1)	15 (46.9)	1	17 (53.1)	15 (46.9)	1	26 (81.3)	6 (18.8)	**0.039**
		**Total**	**17,507 (66.7)**	**8699 (33.2)**	**29 (0.1)**		**15,206 (58.0)**	**11,029 (42.0)**		**19,081 (72.7)**	**7154 (27.3)**		**21,294 (81.2)**	**4917 (18.8)**	

Table 3 shows the frequency distribution of ESAVIs; the numbers in parentheses show the row percentages for each variable (gender, comorbidities, history of COVID-19 infection, and age group). * The *p*-value was obtained by Chi-Square test.

**Table 4 vaccines-10-01047-t004:** Self-reported ESAVIs after receiving the first and second dose of any of the available COVID-19 vaccines in Ecuador (*n* = 6645).

Self-Reported ESAVIs after COVID-19 Vaccination		First Dose		Second Dose		*p* Value *
		n	(%)	n	(%)	
**Time of ESAVIs onset**	Minutes	490	7.4%	210	3.2%	**<0.001**
	Hours	3317	49.8%	1325	19.9%	
	Days	921	13.8%	463	7.0%	
	Weeks	69	1.0%	0,0	0.0%	
	No side effects	1857	27.9%	4656	70.0%	
	**Total**	**6654**	**100%**	**6654**	**100%**	
**Mild self-reported ESAVIs**	Joint pain	1187	7.0%	437	7.4%	0
	Headache	2570	15.2%	939	15.9%	
	Muscle pain	2143	12.6%	761	12.9%	
	Pain or swelling at the injection site	3286	19.4%	1219	20.7%	
	Shaking chills	1659	9.8%	446	7.6%	
	Fatigue or tiredness	2461	14.5%	847	14.4%	
	Mild/low grade fever (37.1–38 °C)	1126	6.6%	345	5.8%	
	General Discomfort	2029	12.0%	770	13.1%	
	Gastrointestinal discomfort	413	2.4%	116	2.0%	
	Urinary discomfort	82	0.5%	20	0.3%	
	**Total reports**	**16,956**	**100%**	**5900**	**100%**	
**Moderate self-reported ESAVIs**	Diarrhea	296	13.1%	108	14.7%	0
	Skin rash	119	5.3%	40	5.4%	
	Fever (>38°)	654	29.1%	175	23.8%	
	Swollen glands	279	12.4%	145	19.7%	
	Nausea	566	25.1%	165	22.4%	
	Pruritus	237	10.5%	64	8.7%	
	Vomit	100	4.4%	39	5.3%	
	**Total reports**	**2251**	**100%**	**736**	**100%**	
**Severe self-reported ESAVIs**	Anaphylaxis	8	2.7%	1	0.7	0
	Tachycardia	176	59.9%	81	57.9%	
	Guillain Barre sd.	7	2.4%	3	2.1%	
	Facial swelling	20	6.8%	6	4.3%	
	High blood pressure	55	18.7%	32	22.9%	
	Transverse myelitis	2	0.7%	0	0.0%	
	Facial paralysis	6	2.0%	5	3.6%	
	Coagulation disorders	20	6.8%	12	8.6%	
	**Total reports**	**294**	**100%**	**140**	**100%**	
**Measures to mitigate self-reported ESAVIs**	Only wait	2295	34.5%	883	13.3%	**<0.001**
	Taking medication	2929	44.0%	1151	17.3%	
	Visiting a physician	319	4.8%	126	1.9%	
	Hospitalization	23	0.3%	10	0.2%	
	No response	1088	16.4%	4484	67.4%	
	**Total**	**6654**	**100%**	**6654**	**100%**	

* The *p*-value was obtained by Chi-Square test.

**Table 5 vaccines-10-01047-t005:** Frequency of self-reported adverse events by age group among 6654 patients.

	Symptoms	Age (Years)																		
		10 to 20		20 to 30		30 to 40		40 to 50		50 to 60		60 to 70		70 to 80		80 to 90		90 to 100		Total
		*n*	(%)	*n*	(%)	*n*	(%)	*n*	(%)	*n*	(%)	*n*	(%)	*n*	(%)	*n*	(%)	*n*	(%)	*n*
**Mild self-reported ESAVIs**	**Joint pain**	15	0.9%	377	23.3%	541	33.4%	359	22.2%	218	13.5%	89	5.5%	16	1.0%	2	0.1%	2	0.1%	1619
	**Headache**	90	2.6%	907	25.9%	1067	30.5%	721	20.6%	476	13.6%	192	5.5%	35	1.0%	13	0.4%	1	0.0%	3502
	**Muscle pain**	59	2.0%	763	26.3%	899	31.0%	587	20.3%	389	13.4%	157	5.4%	32	1.1%	9	0.3%	2	0.1%	2897
	**Pain or swelling at the injection site**	100	2.2%	1162	25.8%	1388	30.8%	959	21.3%	574	12.8%	259	5.8%	42	0.9%	11	0.2%	5	0.1%	4500
	**Fatigue or tiredness**	100	3.0%	949	28.7%	1045	31.6%	645	19.5%	358	10.8%	140	4.2%	45	1.4%	20	0.6%	0	0.0%	3302
	**Shaking chills**	35	1.7%	605	28.8%	695	33.1%	425	20.2%	244	11.6%	77	3.7%	16	0.8%	2	0.1%	1	0.0%	2100
	**Mild/low grade fever (37.1–38 °C)**	16	1.1%	453	30.8%	488	33.2%	288	19.6%	153	10.4%	57	3.9%	12	0.8%	2	0.1%	0	0.0%	1469
	**General Discomfort**	67	2.4%	785	28.1%	884	31.6%	564	20.2%	320	11.5%	130	4.7%	31	1.1%	11	0.4%	2	0.1%	2794
	**Gastrointestinal Discomfort**	5	0.9%	133	25.1%	148	28.0%	133	25.1%	68	12.9%	35	6.6%	5	0.9%	1	0.2%	1	0.2%	529
	**Urinary Discomfort**	0	0.0%	29	28.4%	33	32.4%	23	22.5%	12	11.8%	4	3.9%	0	0.0%	0	0.0%	1	1.0%	102
**Moderate self-reported ESAVIs**	**Diarrhea**	8	2.0%	84	20.8%	121	30.0%	95	23.5%	52	12.9%	33	8.2%	8	2.0%	2	0.5%	1	0.2%	404
	**Skin rash**	5	3.1%	39	24.5%	52	32.7%	28	17.6%	21	13.2%	13	8.2%	1	0.6%	0	0.0%	0	0.0%	159
	**Fever (>38 °C)**	16	1.9%	261	31.5%	245	29.6%	164	19.8%	101	12.2%	33	4.0%	7	0.8%	2	0.2%	0	0.0%	829
	**Swollen Glands**	10	2.4%	119	28.1%	123	29.0%	99	23.3%	58	13.7%	15	3.5%	0	0.0%	0	0.0%	0	0.0%	424
	**Nausea**	18	2.5%	218	29.8%	225	30.8%	145	19.8%	86	11.8%	27	3.7%	10	1.4%	1	0.1%	1	0.1%	731
	**Pruritus (Itching)**	12	4.0%	86	28.6%	74	24.6%	77	25.6%	36	12.0%	16	5.3%	0	0.0%	0	0.0%	0	0.0%	301
	**Vomit**	4	2.9%	50	36.0%	38	27.3%	23	16.5%	18	12.9%	6	4.3%	0	0.0%	0	0.0%	0	0.0%	139
**Severe self-reported ESAVIs**	**Anaphylaxis**	1	11.1%	4	44.4%	2	22.2%	1	11.1%	1	11.1%	0	0.0%	0	0.0%	0	0.0%	0	0.0%	9
	**Tachycardia**	6	2.3%	61	23.7%	59	23.0%	67	26.1%	37	14.4%	17	6.6%	9	3.5%	1	0.4%	0	0.0%	257
	**Guillain Barre sd.**	0	0.0%	1	10.0%	3	30.0%	4	40.0%	1	10.0%	1	10.0%	0	0.0%	0	0.0%	0	0.0%	10
	**Facial swelling**	1	3.8%	3	11.5%	11	42.3%	3	11.5%	6	23.1%	1	3.8%	1	3.8%	0	0.0%	0	0.0%	26
	**High Blood Pressure**	0	0.0%	9	10.3%	16	18.4%	25	28.7%	22	25.3%	12	13.8%	2	2.3%	1	1.1%	0	0.0%	87
	**Transverse Myelitis**	0	0.0%	2	100.0%	0	0.0%	0	0.0%	0	0.0%	0	0.0%	0	0.0%	0	0.0%	0	0.0%	2
	**Facial Paralysis**	0	0.0%	2	18.2%	2	18.2%	4	36.4%	3	27.3%	0	0.0%	0	0.0%	0	0.0%	0	0.0%	11
	**Coagulation Disorders**	0	0.0%	5	15.6%	9	28.1%	12	37.5%	2	6.3%	2	6.3%	2	6.3%	0	0.0%	0	0.0%	32
	**Total**	**568**	**2.2%**	**7107**	**27.1%**	**8168**	**31.1%**	**5451**	**20.8%**	**3256**	**12.4%**	**1316**	**5.0%**	**274**	**1.0%**	**78**	**0.3%**	**17**	**0.1%**	**26,235**

**Table 6 vaccines-10-01047-t006:** Self-reported ESAVIs after the first dose of any of the available vaccines.

	AstraZeneca		Pfizer/BioNTech		Sinovac		*p* Value *
Symptoms	(n)	(%)	(n)	(%)	(n)	(%)	
Tachycardia	97	0.6%	40	0.5%	39	0.7%	**<0.001**
Diarrhea	147	1.0%	62	0.8%	87	1.5%	**<0.001**
Joint Pain	862	5.7%	216	2.6%	109	1.9%	**<0.001**
Headache	1372	9.1%	627	7.7%	571	9.7%	**<0.001**
Muscle Pain	1338	8.9%	507	6.2%	298	5.1%	**<0.001**
Pain Or Swelling at The Injection Site	1273	8.5%	1348	16.5%	665	11.3%	**<0.001**
Skin Rash	56	0.4%	38	0.5%	25	0.4%	**0.002**
Shaking chills	1236	8.2%	250	3.1%	173	2.9%	**<0.001**
Fatigue Or Tiredness	1266	8.4%	620	7.6%	575	9.8%	**<0.001**
Mild/low grade fever (37.1–38 °C)	857	5.7%	194	2.4%	75	1.3%	**<0.001**
Fever (>38 °C)	525	3.5%	88	1.1%	41	0.7%	**<0.001**
Guillain Barre sd.	4	0.0%	3	0.0%	0	0.0%	0.705
Facial Swelling	9	0.1%	8	0.1%	3	0.1%	0.212
High Blood Pressure	25	0.2%	23	0.3%	7	0.1%	**0.004**
Swollen Glands	115	0.8%	105	1.3%	59	1.0%	**<0.001**
General Discomfort	1297	8.6%	427	5.2%	305	5.2%	**<0.001**
Transverse Myelitis	1	0.0%	1	0.0%	0	0.0%	1
Gastrointestinal Discomfort	231	1.5%	81	1.0%	101	1.7%	**<0.001**
Urinary Discomfort	58	0.4%	13	0.2%	11	0.2%	**<0.001**
Nausea	317	2.1%	116	1.4%	133	2.3%	**<0.001**
Facial Paralysis	2	0.0%	1	0.0%	3	0.1%	0.606
Pruritus (Itching)	109	0.7%	74	0.9%	54	0.9%	**<0.001**
Coagulation Disorders	14	0.1%	4	0.0%	2	0.0%	**0.002**
Vomit	56	0.4%	20	0.2%	24	0.4%	**<0.001**
Anaphylaxis	2	0.0%	2	0.0%	4	0.1%	0.607
**Total**	**11,269**	**100%**	**4868**	**100%**	**3364**	**100%**	

* The *p*-value was obtained by Chi-Square test.

**Table 7 vaccines-10-01047-t007:** Self-reported ESAVIs after receiving the second dose of any of the available vaccines.

Symptoms	AstraZeneca		Pfizer/BioNTech		Sinovac		*p* Value
	n	%	n	%	n	%	
TachycardiaDiarrhea	9	0.4%	61	0.7%	11	0.6%	**<0.001**
	27	1.3%	58	0.7%	23	1.3%	**<0.001**
Joint Pain	58	2.7%	336	3.9%	43	2.4%	**<0.001**
Headache	185	8.7%	620	7.2%	134	7.6%	**<0.001**
Muscle Pain	101	4.8%	577	6.7%	83	4.7%	**<0.001**
Pain Or Swelling at The Injection Site	164	7.7%	887	10.3%	168	9.5%	**<0.001**
Skin Rash	6	0.3%	27	0.3%	7	0.4%	**0.003**
Shaking chills	74	3.5%	342	4.0%	30	1.7%	**<0.001**
Fatigue Or Tiredness	152	7.2%	575	6.7%	120	6.8%	**<0.001**
Mild/low grade fever (37.1–38 °C)	48	2.3%	283	3.3%	14	0.8%	**<0.001**
Fever (>38 °C)	27	1.3%	142	1.6%	6	0.3%	**<0.001**
Guillain Barre sd.	0	0.0%	2	0.0%	1	0.1%	0.563
Facial Swelling	1	0.0%	4	0.0%	1	0.1%	0.368
High Blood Pressure	8	0.4%	22	0.3%	2	0.1%	**<0.001**
Swollen Glands	7	0.3%	124	1.4%	14	0.8%	**<0.001**
General Discomfort	123	5.8%	562	6.5%	85	4.8%	**<0.001**
Transverse Myelitis	0	0.0%	0	0.0%	0	0.0%	N/D
Gastrointestinal Discomfort	21	1.0%	69	0.8%	26	1.5%	**<0.001**
Urinary Discomfort	4	0.2%	15	0.2%	1	0.1%	0.472
Nausea	35	1.7%	105	1.2%	25	1.4%	**<0.001**
Facial Paralysis	1	0.0%	2	0.0%	2	0.1%	1
Pruritus (Itching)	10	0.5%	43	0.5%	11	0.6%	**<0.001**
Coagulation Disorders	7	0.3%	5	0.1%	0	0.0%	0.779
Vomit	5	0.2%	32	0.4%	2	0.1%	**0.003**
Anaphylaxis	1	0.0%	0	0.0%	0	0.0%	0.607
**Total**	**1074**	**100%**	**4893**	**100%**	**809**	**100%**	

## Data Availability

The dataset with the total responses can be obtained from the authors upon reasonable request. The questionnaire summary and its results are included in this article.

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
