# Peer review of "A Comparative Analysis of a Self-Reported Adverse Events Analysis after Receiving One of the Available SARS-CoV-2 Vaccine Schemes in Ecuador"

_vaccines, 2022, doi:10.3390/vaccines10071047_

Round 1
Reviewer 1 Report
General comments:
· The study, showing that the self-reported adverse events attributed to COVID-19 vaccines are mostly mild and transitory, can support the safety of vaccination, important to reduce hesitancy in vaccination in Latin America.
· The manuscript is too verbose and we think it could be synthesized.
· The dataset is valid and large, though the results are not always well organised. As the size of the sample, the analytic approach could be improved with further statistical analysis.
· The aim to describe the management of the adverse reactions (found in the introduction) is not really discussed in the text and not mentioned in the conclusion section.
· The paper should be edited for English grammar and usage.
· Please use “point” as decimal separator instead of the comma.
The following suggestions for major edits may help to improve it.
FIGURES AND TABLES
· Figures and Tables could be improved by using different type or image formats. Specific comments are below.
INTRODUCTION
· Abbreviations such as FDA, CDC, WHO must be written in full the first time with acronyms in brackets; instead, acronyms as “VITT” that is used a single time all over the text is not necessary.
· Dealing with serious ESAVIs already described in literature, it would be better to citing examples of each/most vaccines type discussed in the article ( e.g. In reference 12 please specify which vaccine is taken in consideration).
· The aims stated are “to describe the self-reported vaccine adverse events after the application of the available COVID-19 vaccines in Ecuador in addition to the management used to treat reactions”. This latter is not clearly explained and no real management of reactions appeared to be objective of study.
MATERIALS AND METHODS
· “Independent 32 items-self reporting online questionnaire”: it is not clear to me the meaning of independent in this definition.
· Formulas lack detailed legends for each variable (e.g. x; Sqrt). Moreover, a reference is needed in order to explain the calculations rational.
· In Survey development and measures, please add a reference for previous COVID-19 surveys which items were partially used.
STATISTICAL ANALYSIS
· Please specify the type of Chi Square test used. Please explain how P-values reported in Table 3, 4, 5 and 6 were calculated.
· T-Test used is not described in this section. Please explain it.
· As the size of the sample taken in consideration, the analytic approach could be improved with further statistical analysis.
RESULTS
· Age (Table 2): it would be interesting to know which is the age distribution in Ecuadorian population compared to the sample’s one. Moreover, the age distribution by vaccine type may be useful to better understand your results (Has the AstraZeneca vaccine been given mainly to young people?).
· Comorbidities (Table 2): it would be better to know if 4,107 individuals (difference between 6,654 and 2,547) declared to be healthy with no comorbidities or if they did not give a response. Please specify it in the Table/text.
· General characteristics of Self-reported ESAVIs: percentages listed in the first lines of paragraph 3.2.1. (64%, 20.1%, 16.91%) are not supported by data in Table 3. Table 3 is overfilled and does not give clear information. Symptoms described are different from the ones listed in Table 4; ; we suggest to use the same list of symptoms (mild/moderate(severe) in both tables. Table 3 needs a legend with the explanation of what is in brackets (e.g. line/column percentages). Table 3 title is not clear as demographic characteristics are only gender and age.
· Figure 1 is never recall in the text. The percentage are not understandable (which denominator did you use in percentage calculation?). Instead of this figure it would be better to have a figure or a table to explain the number of people reported ESAVIs and their characteristics.
· The total number of self-reported ESAVIs (for the first dose of vaccine) does not seem to be correct: 16,933 (mild) + 2,251 (moderate) + 2,94 (severe) = 19,478 ESAVIs.
· Table 4 contains both frequencies of individuals (e.g. time of ESAVIs onset and measure to mitigate ESAVIs with total number 6,654) and frequencies of reported ESAVIs. More clarity would be necessary. Numbers of individuals in section “time of ESAVIs onset” are not coherent with ones of “measures to mitigate self-reported ESAVIs”. Is it due to blanket answers in the questionnaire?
Finally, in the text, no comment about management of reactions is mentioned (even if it appears to be one of the two aims of your work).
· Figures 2 and 3 are very difficult to read. We suggest to replace Figure 2 and 3 with a Table showing ESAVIs for class of age.
· Dealing with data from Tables 5 and 6, we think that a relevant information -lacking in this work- should be the incidence of symptoms listed on vaccinated population with each type of vaccine. Moreover, it would be interesting to compare ESAVIs consequent to the three different type of vaccines administered to homogeneous groups of age (e.g. young people received AstraZeneca suffered more of headache respect to young ones received Pfizer?)
· Finally, your sample is large enough to estimate the correlation (using suitable statistical models) between ESAVIs and vaccine types/individuals’ characteristics. This could give more foundation to the analysis presented in the text.
DISCUSSION
· In the second paragraph it is stated that female sex and age <50 years had more ESAVIs compared to male sex and older population. Those evidences should be proved by p-values considering the significance respect to the population composition (75% were <50 years and 60% were female sex).
· In the discussion of the effects cause by AstraZeneca e Pfizer vaccines (page 14) sentences are unclear (especially in the comparison with other studies) with a not fluid speech. Some sentences are not understandable for grammatical errors.
· Regarding the paragraph on Sinovaq, symptoms described appear to be equal in all references recalled; in that case we suggest to state the accordance of your results with them more concisely.
· In the last paragraph please rearrange the sentence from “on the other hand” to “Sinopharm” as it is too long and difficult to understand.
CONCLUSION
We invite you to revise the conclusion section after the revisions suggested.
Conclusion should contain, concisely, the main messages supported by your analysis and coherent with your aims.
Author Response
To: Editor in Chief Vaccines Journal (ISSN 2076-393X)
Title: A comparative analysis of a self-reported adverse events analysis after receiving one of the available SARS-CoV-2 vaccine schemes in Ecuador.
Manuscript ID :vaccines-1748086
Dear Editor and reviewers, thank you very much for your effort in observing our manuscript and offering us some comments intended to improve our manuscript.
We have completed a full revision which includes answers to all your comments and suggestions. All changes are highlighted in red within the main manuscript and this point-by-point letter.
Reviewer 1
General comments:
- The study, showing that the self-reported adverse events attributed to COVID-19 vaccines are mostly mild and transitory, can support the safety of vaccination, important to reduce hesitancy in vaccination in Latin America.
Thanks for your comment
- The manuscript is too verbose and we think it could be synthesized.
Thanks a lot, we tried to synthetize it when possible and we use less wordy sentences thought-out the entire manuscript
- The dataset is valid and large, though the results are not always well organised. As the size of the sample, the analytic approach could be improved with further statistical analysis.
Thanks for your observation, we have done so and simplify data description as much as possible
- The aim to describe the management of the adverse reactions (found in the introduction) is not really discussed in the text and not mentioned in the conclusion section.
We have included the discussion and we have improved our conclusions
- The paper should be edited for English grammar and usage.
We have done so, thanks for your suggestions
- Please use “point” as decimal separator instead of the comma.
Great, we have done so
The following suggestions for major edits may help to improve it.
FIGURES AND TABLES
- Figures and Tables could be improved by using different type or image formats. Specific comments are below.
- We have improved figures and tables throughout the entire manuscript.
INTRODUCTION
- Abbreviations such as FDA, CDC, WHO must be written in full the first time with acronyms in brackets; instead, acronyms as “VITT” that is used a single time all over the text is not necessary.
Thanks, we have added the meaning of the abbreviation.
- Dealing with serious ESAVIs already described in literature, it would be better to citing examples of each/most vaccines type discussed in the article (e.g. In reference 12 please specify which vaccine is taken in consideration).
We have added the serious ESAVIs for the vaccines used in the article (BNT162b2, CoronaVac and ChAdOx1).
- The aims stated are “to describe the self-reported vaccine adverse events after the application of the available COVID-19 vaccines in Ecuador in addition to the management used to treat reactions”. This latter is not clearly explained and no real management of reactions appeared to be objective of study.
We corrected the aims.
MATERIALS AND METHODS
- “Independent 32 items-self reporting online questionnaire”: it is not clear to me the meaning of independent in this definition.
It was an unnecessary term, we removed "independent" from the paragraph.
- Formulas lack detailed legends for each variable (e.g. x; Sqrt). Moreover, a reference is needed in order to explain the calculations rational.
We add additional information and the support references to better explain our calculations.
- In Survey development and measures, please add a reference for previous COVID-19 surveys which items were partially used.
We add the references from which we started for the development of our instrument.
STATISTICAL ANALYSIS
- Please specify the type of Chi Square test used. Please explain how P-values reported in Table 3, 4, 5 and 6 were calculated.
We add information to specify the type of Chi-Square test used and describe the methods used for the p-values calculation.
- T-Test used is not described in this section. Please explain it.
We add information about the t test and the contrast of hypotheses used.
- As the size of the sample taken in consideration, the analytic approach could be improved with further statistical analysis.
The size of the study sample is important. In general, we have used the t-test and Chi-Square tests, having obtained about 11,927 responses, however, when creating strata of valid responses (less than 7,000) around the adverse effects, the size per group was small, which is why that the use of other tests was not convenient, and they would not have contributed conclusions to this study.
RESULTS
- Age (Table 2): it would be interesting to know which is the age distribution in Ecuadorian population compared to the sample’s one. Moreover, the age distribution by vaccine type may be useful to better understand your results (Has the AstraZeneca vaccine been given mainly to young people?).
We added a figure (Figure 1) to explain the distribution of COVID-19 vaccines in our participants relative to the Ecuadorian population.
- Comorbidities (Table 2): it would be better to know if 4,107 individuals (difference between 6,654 and 2,547) declared to be healthy with no comorbidities or if they did not give a response. Please specify it in the Table/text.
We added information within the text.
- General characteristics of Self-reported ESAVIs: percentages listed in the first lines of paragraph 3.2.1. (64%, 20.1%, 16.91%) are not supported by data in Table 3. Table 3 is overfilled and does not give clear information. Symptoms described are different from the ones listed in Table 4; ; we suggest to use the same list of symptoms (mild/moderate(severe) in both tables. Table 3 needs a legend with the explanation of what is in brackets (e.g. line/column percentages). Table 3 title is not clear as demographic characteristics are only gender and age.
We have amended the data in the text of section 3.2.1. We unified the list of symptoms described in tables 3 and 4. We added the caption for table 3. We corrected the title of table 3.
- Figure 1 is never recall in the text. The percentage are not understandable (which denominator did you use in percentage calculation?). Instead of this figure it would be better to have a figure or a table to explain the number of people reported ESAVIs and their characteristics.
We decided to remove Figure 1 since the ESAVIS distribution data by sex is found in the "gender" column of Table 3.
- The total number of self-reported ESAVIs (for the first dose of vaccine) does not seem to be correct: 16,933 (mild) + 2,251 (moderate) + 2,94 (severe) = 19,478 ESAVIs.
We correct the total number of the adverse effects of the first dose.
- Table 4 contains both frequencies of individuals (e.g. time of ESAVIs onset and measure to mitigate ESAVIs with total number 6,654) and frequencies of reported ESAVIs. More clarity would be necessary. Numbers of individuals in section “time of ESAVIs onset” are not coherent with ones of “measures to mitigate self-reported ESAVIs”. Is it due to blanket answers in the questionnaire?
Discrepancies between the " time of ESAVIs onset " and " measures to mitigate self-reported ESAVIs " sections are due to questionnaire responses. We added data in the measures to mitigate ESAVIS section of table 4, to eliminate discrepancies and improve understanding
- Finally, in the text, no comment about management of reactions is mentioned (even if it appears to be one of the two aims of your work).
We correct the aims.
- Figures 2 and 3 are very difficult to read. We suggest to replace Figure 2 and 3 with a Table showing ESAVIs for class of age.
We remove figures 2 and 3, and in their place, we add a new table (Table 5)
- Dealing with data from Tables 5 and 6, we think that a relevant information -lacking in this work- should be the incidence of symptoms listed on vaccinated population with each type of vaccine. Moreover, it would be interesting to compare ESAVIs consequent to the three different type of vaccines administered to homogeneous groups of age (e.g. young people received AstraZeneca suffered more of headache respect to young ones received Pfizer?)
We add a new figure (figure n) to describe the relationship between ESAVIs / type of vaccine / age ranges. In addition, when solving the previous comments, tables 5 and 6 became tables 6 and 7.
- Finally, your sample is large enough to estimate the correlation (using suitable statistical models) between ESAVIs and vaccine types/individuals’ characteristics. This could give more foundation to the analysis presented in the text.
thanks for your suggestions, we have included the analysis of correlational calculations throughout the document.
DISCUSSION
- In the second paragraph it is stated that female sex and age <50 years had more ESAVIs compared to male sex and older population. Those evidences should be proved by p-values considering the significance respect to the population composition (75% were <50 years and 60% were female sex).
We clarify information regarding the statistically significant differences that we wanted to discuss.
- In the discussion of the effects cause by AstraZeneca e Pfizer vaccines (page 14) sentences are unclear (especially in the comparison with other studies) with a not fluid speech. Some sentences are not understandable for grammatical errors.
We improve grammar and better explained comparisons between our findings and other studies.
- Regarding the paragraph on Sinovaq, symptoms described appear to be equal in all references recalled; in that case we suggest to state the accordance of your results with them more concisely.
We clarify more precisely the similarities and differences between our findings and those of the references.
- In the last paragraph please rearrange the sentence from “on the other hand” to “Sinopharm” as it is too long and difficult to understand.
CONCLUSION
We invite you to revise the conclusion section after the revisions suggested.
Conclusion should contain, concisely, the main messages supported by your analysis and coherent with your aims.
Thanks so much for this, we have improved our conclusion section
Reviewer 2 Report
In this manuscript, Esteban Ortiz-Prado and colleagues reported the ESAVIs after one or two COVID-19 vaccine doses in Latin America. They found women, young adults, participants without COVID-19 infection history, participants after 238 the first vaccine dose, and AstraZeneca users developed ESAVIs more frequently. This work is very important and I suggest publish this work with minor revision.
My minor points are:
1. The table 1 summary the main characteristics of the different mechanisms of action used by different COVID-19 vaccines. But it is an incomplete summary. Some approved vaccines are not included, such as the protein vaccine ZF2001 (PMID:.35507481)
2. It is better to add some backgrounds on the clinical result of AstraZeneca, Pfizer, and Sinovac vaccines in the Introduction.
3. The abbreviation that first appeared should write the full name before it, please double check.
Author Response
In this manuscript, Esteban Ortiz-Prado and colleagues reported the ESAVIs after one or two COVID-19 vaccine doses in Latin America. They found women, young adults, participants without COVID-19 infection history, participants after 238 the first vaccine dose, and AstraZeneca users developed ESAVIs more frequently. This work is very important and I suggest publish this work with minor revision.
Thanks so much for your comments, we really appreciate it
My minor points are:
- The table 1 summary the main characteristics of the different mechanisms of action used by different COVID-19 vaccines. But it is an incomplete summary. Some approved vaccines are not included, such as the protein vaccine ZF2001 (PMID:.35507481)
We add the vaccines BBV152 (Covaxin), ZF2001 (Zifivax), AD5-nCOV (Convidecia), CIGB-66 (ABDALA) and FINLAY-FR-2 (Soberana 02) to Table 1.
- It is better to add some backgrounds on the clinical result of AstraZeneca, Pfizer, and Sinovac vaccines in the Introduction.
We add a background of these vaccines.
- The abbreviation that first appeared should write the full name before it, please double check.
We write de full name of the abbreviations.
Reviewer 3 Report
Major Comments
Τhe manuscript “A comparative analysis of a self-reported adverse events analysis after receiving one of the available SARS-CoV-2 vaccine schemes in Ecuador” is, at least according to the authors, “the first study in Latin America examining self-reported ESAVIs after one or two COVID-19 vaccine doses”; in this respect, its publication would be of particulate interest to the readers of Vaccines. However, there are some issues that should be successfully addressed before the manuscript can be accepted for publication. Among them, the most serious ones are related with the Discussion part, which should be re-written in a clearer and more concise way (e.g., the text from line 268 to line 289 is especially difficult to follow -lack of punctuation is an additional problem). Moreover, some specific points should be explicitly clarified in the Discussion part, e.g., have self-reported severe side effects, such as Guillain Barre syndrome, been “officially” diagnosed by authorized medical personnel?? If not, this has to be clearly stated and included among the limitations of the study. On the other hand, at least in my opinion, it would be of interest if the authors could compare the main technical parameters of their work, e.g., study design, sample selection, data management and statistical analysis, with those employed in relevant studies conducted in other countries and referred to in Introduction and/or Discussion of this manuscript.
Minor Comments
Line 27: Please, correct 20,01% and 16,91% to 20.01% and 16.91%, respectively (please, note that “,” is used instead of “.” in many numbers that appear throughout the text.
Line 28: Please, provide the full term abbreviated as ESAVIs (events supposedly attributable to vaccination or immunization)
Table 1 (a): There is no footnote explaining what the symbol * in (US)*, (UK)*, (China)* stands for.
Table 1 (b): 5 * 1010 -is this 5 x 1010?
Lines 98-99, “…the sample size n and margin error Ε are given in the following formula”: Would it be possible for the authors to provide a reference?
Lines 119-120, “The full survey instrument is available in the Additional file 1”: I could not have access to this file -perhaps I have missed something during the documents’ downloading?
Line 163, “Most responders were aged between 21 to 50 years (73.9%) were employed (79.60%);” the syntax here seems to be incomplete.
Table 3, second column (“Female”), 9th number (220 (66.9)): presumably, the correct numbers here are “2,200 (66.9)”??
Figure 1, Y-axis: it is unclear what is written in the parentheses after “Fever”
Figure 1 & Table 4: “oC” instead of “o”
Lines 187-188: “Self-reported ESAVIs were higher among young adults (55.8%) and adults (34.5%) compared to older adults (7.6%), …”: Please, specify the age-ranges for “young adults”, “adults” and “older adults”.
Author Response
Major Comments
Τhe manuscript “A comparative analysis of a self-reported adverse events analysis after receiving one of the available SARS-CoV-2 vaccine schemes in Ecuador” is, at least according to the authors, “the first study in Latin America examining self-reported ESAVIs after one or two COVID-19 vaccine doses”; in this respect, its publication would be of particulate interest to the readers of Vaccines. However, there are some issues that should be successfully addressed before the manuscript can be accepted for publication.
Thanks for your comments, we truly appreciate your time reading it
Among them, the most serious ones are related with the Discussion part, which should be re-written in a clearer and more concise way (e.g., the text from line 268 to line 289 is especially difficult to follow -lack of punctuation is an additional problem). Moreover, some specific points should be explicitly clarified in the Discussion part, e.g., have self-reported severe side effects, such as Guillain Barre syndrome, been “officially” diagnosed by authorized medical personnel?? If not, this has to be clearly stated and included among the limitations of the study.
Thanks so much for pointing this out, we have amended the entire manuscript and reviewed for clarity and readability
On the other hand, at least in my opinion, it would be of interest if the authors could compare the main technical parameters of their work, e.g., study design, sample selection, data management and statistical analysis, with those employed in relevant studies conducted in other countries and referred to in Introduction and/or Discussion of this manuscript.
We have added some comments regarding other instruments, beside, the wording of the discussion section was improved. Specifically, the text between lines 268 - 289 was restructured. Information was added regarding the limitations of the study's findings.
Minor Comments
Line 27: Please, correct 20,01% and 16,91% to 20.01% and 16.91%, respectively (please, note that “,” is used instead of “.” in many numbers that appear throughout the text.
We have added your suggestions
Line 28: Please, provide the full term abbreviated as ESAVIs (events supposedly attributable to vaccination or immunization)
We added the full term
Table 1 (a): There is no footnote explaining what the symbol * in (US)*, (UK)*, (China)* stands for.
It was a mistake; we erase de symbol *
Table 1 (b): 5 * 1010 -is this 5 x 1010?
We changed 5 * 1010 to for 5 x 1010
Lines 98-99, “…the sample size n and margin error Ε are given in the following formula”: Would it be possible for the authors to provide a reference?
We have added a proper reference
Lines 119-120, “The full survey instrument is available in the Additional file 1”: I could not have access to this file -perhaps I have missed something during the documents’ downloading?
We update and make sure to upload the additional files
Line 163, “Most responders were aged between 21 to 50 years (73.9%) were employed (79.60%);” the syntax here seems to be incomplete.
We have completed the main point of the idea
Table 3, second column (“Female”), 9th number (220 (66.9)): presumably, the correct numbers here are “2,200 (66.9)”??
It was a mistake, the correct number is 2,209, we add it.
Figure 1, Y-axis: it is unclear what is written in the parentheses after “Fever”
We replace figure 1 with table 5
Figure 1 & Table 4: “oC” instead of “o”
We replace figure 1 with table 5 and correct the symbols
Lines 187-188: “Self-reported ESAVIs were higher among young adults (55.8%) and adults (34.5%) compared to older adults (7.6%), …”: Please, specify the age-ranges for “young adults”, “adults” and “older adults”.
We specify the ranges used for the frequencies
Round 2
Reviewer 1 Report
General comments:
· The manuscript is too verbose, sentences are too long (check the punctuation) please synthesize.
· The paper should be further edited for English grammar and usage.
· Please use “point” as decimal separator instead of the comma (page 9, page 14 etc).
The following suggestions for minor edits may help to improve it.
INTRODUCTION
· “The Pfizer Phase III….adolescents aged 12-15” is difficult to understand, please correct the language and use shorter sentences.
· “In the case of the BNT…[19]” in this period punctuation is lacking.
MATERIALS AND METHODS
· Formula lacks legend for variable x.
RESULTS
· Table 3 total results for mild, moderate and severe ESAVIs should be added.
· Table 5 is not easy to read and very detailed but maybe it would be more interesting to show the frequencies distributions of ESAVIs (mild, moderate, severe) for classes of age (3 or 4 classes of age) and type of vaccine. Please try to see if results are worth to be described.
· Table 4 and Table 6-7 total ESAVIs should be the same. Please clarify if there were differences (in table 4 total ESAVIs after the first dose is 19478; in Table 6 is 19501 for example; for second dose the total is 6757 versus 6776). Please correct also in the test.
Author Response
Dear Reviewer, thanks for your observations, we really appreciate them since they will make our manuscript more readable and easier to understand.
Please find attached our responses
General comments:
- The manuscript is too verbose, sentences are too long (check the punctuation) please synthesize.
We have eliminated several lines and sentences, and we tried to synthesize our results as much as possible without losing context.
- The paper should be further edited for English grammar and usage.
We have reviewed the entire manuscript for mistakes and readability
- Please use “point” as a decimal separator instead of the comma (page 9, page 14, etc).
We have changed the separator symbol to the point where required.
The following suggestions for minor edits may help to improve it.
INTRODUCTION
- “The Pfizer Phase III….adolescents aged 12-15” is difficult to understand, please correct the language and use shorter sentences.
We corrected and improved the wording in the requested sentence
- “In the case of the BNT…[19]” in this period punctuation is lacking.
We added the missing punctuation marks in the requested sentence
MATERIALS AND METHODS
- Formula lacks legend for variable x.
We added the information.
RESULTS
- Table 3 total results for mild, moderate, and severe ESAVIs should be added.
We added the data
- Table 5 is not easy to read and very detailed but maybe it would be more interesting to show the frequency distributions of ESAVIs (mild, moderate, severe) for classes of age (3 or 4 classes of age) and type of vaccine. Please try to see if the results are worth to be described.
We have improved the format of the table and added the frequency distribution between ESAVI severity (mild, moderate, severe).
We considered that adding an extra analysis regrouping age groups and type of vaccines would incur repeating analyses performed in other sections of the manuscript, such as Table 3 in the case of age groups and severity of ESAVIs and Figure 3 where the relationship between ESAVIs, type of vaccine and age groups was studied.
- Table 4 and Table 6-7 total ESAVIs should be the same. Please clarify if there were differences (in table 4 total ESAVIs after the first dose is 19478; in Table 6 is 19501 for example; for the second dose the total is 6757 versus 6776). Please correct also the test.
There were minimal errors in the calculation of the dose 1 and 2 (n) of mild ESAVIs in Table 4.
The correct total for the entire study of ESAVIs at the first dose is 19501 and 6776 for the second dose. We corrected the wrong values and recalculated percentages and Chi-square test for Table 4.